# Population Genomic Approaches for Weed Science

**DOI:** 10.3390/plants8090354

**Published:** 2019-09-19

**Authors:** Sara L. Martin, Jean-Sebastien Parent, Martin Laforest, Eric Page, Julia M. Kreiner, Tracey James

**Affiliations:** 1Ottawa Research and Development Centre, Agriculture and Agri-Food Canada, Ottawa, ON K1A 0C6, Canada; 2Saint-Jean-sur-Richelieu Research and Development Centre, Agriculture and Agri-Food Canada, Saint-Jean-sur-Richelieu, QC J3B 3E6, Canada; 3Harrow Research and Development Centre, Agriculture and Agri-Food Canada, Harrow, ON N0R 1G0, Canada; 4Department of Ecology and Evolutionary Biology, University of Toronto, Toronto, ON M5S 3B2, Canada

**Keywords:** weeds, genomics, plant genome assembly, non-target site resistance, population genomics, genome scans, population genetics

## Abstract

Genomic approaches are opening avenues for understanding all aspects of biological life, especially as they begin to be applied to multiple individuals and populations. However, these approaches typically depend on the availability of a sequenced genome for the species of interest. While the number of genomes being sequenced is exploding, one group that has lagged behind are weeds. Although the power of genomic approaches for weed science has been recognized, what is needed to implement these approaches is unfamiliar to many weed scientists. In this review we attempt to address this problem by providing a primer on genome sequencing and provide examples of how genomics can help answer key questions in weed science such as: (1) Where do agricultural weeds come from; (2) what genes underlie herbicide resistance; and, more speculatively, (3) can we alter weed populations to make them easier to control? This review is intended as an introduction to orient weed scientists who are thinking about initiating genome sequencing projects to better understand weed populations, to highlight recent publications that illustrate the potential for these methods, and to provide direction to key tools and literature that will facilitate the development and execution of weed genomic projects.

## 1. Introduction

Biology is currently in the midst of a revolution caused by the advances in sequencing technology that allow us to examine genomes in detail [1]. Genomic information promises new insights for understanding the biology, evolutionary history, and adaptive potential in ways that were recently out of reach for laboratories studying organisms with genomes larger than model organisms (e.g., *Arabidopsis thaliana* (L.) Heyn. 135 Mb) [2,3,4,5]. Additionally, genomics at the population or species level are now possible in some species and will likely become practical for the majority of organisms in the near term. The huge potential of these advances has been exploited by some disciplines, such as those investigating bacteria [6,7], viruses [8,9] or humans [10,11,12,13], with greater alacrity than others. Notably, however, progress adopting genomic methods has been slow in weed science despite recognition of the power of these methods [14,15].

There are numerous impediments to a greater use of genomics in weed science. One of these elements is the lack of chromosome level reference genome sequences for weeds, as the majority of sequencing efforts have been focused on crops. Genome sequences are foundational for many approaches and the relatively early availability of the human genome sequence [16], model organisms such as *Arabidopsis* [17] and many crop species [18] have been essential to the rapid progress in applying genomic approaches to a wide range of disciplines. This issue has been noted by the weed science community and efforts such as the International Weed Science Consortium have been initiated [14]. However, an additional impediment to using these rapidly developing and expanding set of techniques is a lack of familiarity among weed scientists. As a result, our aim here is to provide a brief primer and introductory “how to guide” and “why would you guide” relevant to weed science. We briefly review *de novo* genome assembly and annotation as these methods are often fundamental for further work. Then we focus on how genomic approaches can be used to answer three key questions: 1) Where do agricultural weeds come from and why are they weedy; 2) what genes underlie herbicide resistance (HR); and, more speculatively, 3) can we alter weed populations to become easier to control? We highlight what resources would be needed for success and provide illustrative examples from both weed science and the broader scientific literature.

## 2. Developing Weed Genome Sequences as a Fundamental Tool

While some genomic approaches do not require a draft genome for the species of interest, the majority of techniques do, or benefit from the availability of at least a rough draft. Sequencing plant genomes is easier than ever before with the decreasing cost of sequencing and the increasing ease with which tools such as genome assembly programs can be installed and used. However, genome assembly remains a challenge that will require a significant investment of time and resources for the majority of weed species [19]. Here we provide a brief outline of how to approach a *de novo* genome sequencing project and provide an initial introduction to the steps required and some tools that could be used as a starting point. We do not attempt to provide a comprehensive list of resources or tools and in every case, there are often numerous alternatives that may be better suited to a particular weed species or easier to install in a specific computing environment. Further, new tools are continuously emerging (and older ones submerging) in this quickly evolving area. Various databases of these tools have been compiled such as omictools.com and bioinformaticssoftwareandtools.co.in. Valuably, a recent review by Jung et al. [20] is comprehensive with recommendations on the computational resources needed to complete these assemblies.

### 2.1. What Is a Draft Genome?

A draft genome of a plant species is a haploid representation of a portion of the total DNA and genes. As such, it is a simplified and limited representation of the total information contained in the genome of the individual sequenced. It will lack information on allelic variation and portions of the genome, especially repetitive elements and material near the centromeres [21]. A draft genome is comprised of a group, often a large group (Table 1), of contigs that vary in size and represent the portions of the genome assembled from overlapping and joining the smaller pieces provided by the sequencing reads and is often presented in a multi-fasta file. These contigs can be assembled into larger fragments, scaffolds. Finally, scaffolds can be assembled, ordered, and oriented into pseudomolecules. At the larger end, pseudomolecules may represent chromosomes, chromosome arms or smaller features such as the chloroplast’s genome. In general, the fewer number of contigs an assembly has the better the assembly is considered. A metric used to compare continuity amongst genomes is the NG50 value. If one ordered all the contigs in an assembly from largest to smallest and added the length of each contig as you went down the list, the NG50 value would be the size of the contig when 50% of the species’ expected genome size was reached [22]. The N50 value is similar and more frequently reported, but as the assembly size is used instead of the expected genome size, it can’t be used to compare different assemblies even within species [22]. Drafts comprised of thousands of contigs can be sufficient for many purposes, including understanding evolutionary relationships among species, acting as a reference for studies of population biology, and for developing molecular identification tools. To understand fine-scale patterns of selection, however, a chromosomal level assembly is more desirable, allowing for the most detailed analysis and inferences that draw on correlated shifts in allele frequencies. In cases where a closely related species has been assembled to the chromosome level and chromosome number is conserved, this may, by assuming synteny (preserved order), be used to position scaffolds into pseudomolecules representing a first guess of what the genome may look like. However, this level of information has rarely been achieved for non-model, non-crop organisms.

### 2.2. Preparing and Assessing Plant Material

Important initial steps to help ensure the success of a project are assessment of the plant material to understand the species’ genome size and composition and carefully considering the starting material including finding lower ploidy individuals or reducing heterozygosity through inbreeding or other genetic manipulations such as creating a doubled haploid.

It is preferable to know the size of the genome before the start of a sequencing project. Several databases have compiled information on the genome size and chromosome counts for plant species (see Rice et al. 2015 for a list of resources). A particularly useful resource for genome size information is the Plant DNA C-value Database (cvalues.science.kew.org) hosted by Kew Royal Botanic Gardens [23]. Similarly, chromosome counts are available from the Index to Plant Chromosome Numbers www.tropicos.org/project/ipcn hosted by the Missouri Botanical Garden and the Chromosome Counts Database ccdb.tau.ac.il [24].

In the absence of information from these sources, or in cases where multiple chromosome counts or DNA contents have been reported, analysis by flow cytometry can determine the DNA content of the material of interest [25,26,27,28]. This is relatively inexpensive and straight forward if you have access to a flow cytometer and can take as little as a week for an experienced laboratory. Fresh tissue is co-chopped in a buffer with the tissue of a species with known DNA content (internal standard), nuclei are stained with a fluorophore such as propidium iodide, and peaks in fluorescence are produced as a result of excitation by the flow cytometer’s laser. Then the position of the sample’s peak and the known standard are determined by analysis of the resulting histogram with appropriate software (e.g., [29]). The DNA content of the samples is then determined using these relative positions and the DNA content of the standard. Generally, at least three individuals should be tested and each analyzed with three technical replicates across three days. This provides the full 2C DNA content of the plant’s nuclei in picograms. The 1C DNA content can then be calculated by dividing this value in half and convert to Mbp by multiplying by 978 Mbp/pg [30]. Difficulties with DNA content determination with flow cytometry typically centre around finding an extraction buffer that allows for the production of narrow peaks and low debris levels (coefficient of variation < 5), including enough nuclei in the sample peak (>1000), finding an appropriate standard, and understanding the data when it is complicated by extra peaks from contamination or endopolyploidy [31,32]. Methods using an external standard should not be used as they are less accurate. This information can be compared to DNA content and chromosome counts for species in the same genus to make educated guesses about the chromosome count for the material of interest, but a conclusive determination of chromosome number requires either the counting of chromosome spreads or the use of more advanced chromosome sorting techniques [33].

Producing chromosome spreads is generally more accessible than chromosome sorting, but requires a significant amount of time, especially if the species’ chromosomes are small or numerous. A pair of highly helpful videos on the technique, produced by the Beck Laboratory are available as an introduction (www.youtube.com/watch?v=iXqni6knH5A&t and www.youtube.com/watch?v=xVV4qBfSQLs&t) [71,72]. Several methods that inhibit spindle formation and increase the accumulation of metaphase cells can be used to facilitate chromosome counts. These include pre-treating material with pressurized nitrous oxide (NO_2_), incubation in ice cold water, or exposing the cells to chemical inhibitors such as 8-hydroxyquinoline or colchicine [73]. For example, for a mitotic preparation, NO_2_ pressurized to 8–10 atm (160 psi) can be applied for several hours to 1 cm long root tips in water using a specially constructed air sealed, iron pressure chamber with the regulator and hoses of the correct composition needed to attach and deliver NO_2_ [74]. The water is then removed and replaced with fresh Carnoy’s fixative for storage at 4 °C. Samples are then washed twice with distilled water and 1x citric buffer respectively. This buffer is replaced with enough 0.3% pectolytic enzyme solution [75] to ensure the material is fully submerged and incubated at 37 °C for 60 min. Digested root tips should form into a cell suspension when they are tapped with dissecting needles on a slide. If clumps form, or cells do not separate, incubation in the enzyme solution should be increased. Once cell suspension has been created, a drop of orcein stain [73] can be added [73], the drop carefully spread, and a coverslip placed on top. Then the slide is heated and squashed between filter paper using thumb pressure, ensuring no slippage. The slide can then be examined with a phase-contrast microscope for the quality of chromosome spread and count. If cytoplasm covers the nuclei then pepsin treatment may be effective [75]. Obtaining a good spread that will allow for certainty in chromosome number will generally take patience and practice.

An alternative to flow cytometry for determining genome size is to complete a k-mer plot of Illumina short read data (Illumina, San Diego, California, USA) [3] using a tool such as KmerGenie (kmergenie.bx.psu.edu) [76] or Jellyfish (www.cbcb.umd.edu/software/jellyfish), however it is preferable to have an estimate independent of the reference read data itself [77] (Figure 1). Following data generation with Jellyfish a script can be written in R to visualize the data or the data can be easily visualized using the website GenomeScope (qb.cshl.edu/genomescope). This data can also provide an indication of heterozygosity and can be used to determine the amount of the genome comprised of repetitive elements using tools such as RepeatExplorer (repeatexplorer.org) [78,79].

While the addition of long-read technology is making the assembly of highly heterozygous and repeat-rich genomes more feasible, genome assembly can be simplified by reducing heterozygosity and repetitive elements. In species that are self-compatible, repeated self-pollination can do both and result in a less redundant and more contiguous assembly [81]. In outcrossing or dioecious species or species with strong inbreeding depression reducing variation can be more difficult, requiring strategies such as repeated full sibling mating. Doubled haploids, generally produced via tissue culture of either male or female gametophytes, can solve this problem by completely eliminating heterozygosity, but are also a significant challenge and investment of time [82,83,84].

For genome sequencing, the 1C DNA content is perhaps the most important piece of information for designing the sequencing strategy, determining the quantity of sequencing required, and providing hints as to the species’ degree of polyploidization or genome size inflation resulting from repetitive element proliferation.

Additional challenges await groups that wish to assemble genomes which have undergone recent or ancient polyploidization, which are notoriously more difficult to assemble, though long reads are making these genomes increasingly tractable. Many successfully sequenced crops fall in this category and specific strategies have been developed to assemble these genomes (reviewed by [85]). However, when a weed species is variable for ploidy the most feasible approach would be to select an individual with the lowest ploidy available for sequencing. However, the conclusions about the species’ population genetics that are drawn from this genome would only be applicable to populations with this cytotype. In any case, a vouchered record of the material used for DNA extraction should be created and submitted to an herbarium to provide documentation of the species that has been sequenced [86].

### 2.3. DNA Extraction

Extraction of DNA of sufficient quality and quantity can be a surprisingly difficult hurdle. Technologies such as Pacific Biosciences’ (PacBio) single molecule real-time (SMRT) sequencing (Pacific Biosciences, Menlo Park, California, USA) and Oxford Nanopore Technologies’ sequencing systems (Oxford Nanopore Technologies, Oxford, UK) require high molecular weight (HMW) DNA at a high concentration (e.g., 10 µgs with an average size of 30–50 kbp for PacBio) [87]. This genomic HMW DNA needs to have little evidence of shearing, be free of contamination from protein, RNA, or polysaccharides and a 260/280 nm absorbance ratio of approximately 1.8–2.0. This is not always simple to achieve and time may need to be devoted to optimizing the DNA extraction protocol.

We have observed that the method of grinding the plant tissue appears to be the most critical step in obtaining HMW DNA with little shearing (Martin, unpublished). While many protocols suggest using bead mills with either ceramic, metal beads and/or sand, using the least time and speed reduces shearing. We have found that grinding tissue in 2 mL tubes with plastic pestles on dry ice, using wide bore tips, minimizing vortexing and pipetting will limit shearing and help ensure recovery of HMW DNA. Commercial kits are convenient and remove contaminants, but often an insufficient amount of DNA is obtained from a single extraction. However, multiple extractions can be pooled and concentrated to obtain the HMW at a sufficient concentration.

When sufficient tissue is available, many genome sequencing projects (e.g., [44,54,88,89]) have found success with variations on the traditional hexadecyltrimethylammonium bromide (CTAB) based method, described by Doyle and Doyle [90]. These methods often use a large quantity (g) of plant tissue ground in liquid nitrogen with a mortar and pestle. Many modifications of this original protocol are available, including Healy et al.’s [91] protocol for plants with large amounts of phenolics and polysaccharides. These compounds can inhibit downstream library preparations and are particularly important to eliminate. If required, further purification can be done with additional ethanol precipitations or magnetic beads (Agilent, Santa Clara, California, USA). For example, a strategy to prepare fragments for sequencing is to shear the DNA into large fragments of 20 kb in size using g-TUBES (Covaris, Woburn, MA, USA) and then selecting fragments of appropriate size with an apparatus such as the Blue Pippin (Sage Science, Beverly, MA, USA). In addition, some laboratories have found specially designed tips, such as Qiagen Genomic Tips (Qiagen, Hilden, Germany) to be helpful during preparation of the samples. Other technologies such as the Short Read Eliminator Kit (Circulomics, Baltimore, MD, USA) can be used to optimize sequencing by removing shorter fragments. Following extraction, DNA integrity and concentration need to be assessed. A variety of tools exist to complete these steps including the Tapestation or Bioanalyzer system (Agilent Genomics) [87]. However, it has been noted that DNA quantities should be measured on a Qubit Fluorometer (Thermo Fisher Scientific, Waltham, Massachusetts, USA) or similar as Nanodrop (Thermo Fisher Scientific) can overestimate quantity [87].

### 2.4. Sequencing Strategies

Assembling a genome using large pieces is much easier than using small pieces. Therefore, the majority of sequencing projects now combine long read (e.g., PacBio or ONT) and short read data. Long reads, which generally average 10 kb or more in length, make assembling plant genomes comparatively easier and general result in a more contiguous assembly. Genome assembly is sensitive to repeated sequences and these can only be resolved if the sequencing technology spans the regions. However, the error rate for long reads maybe as high as 15% and therefore require greater depth (30× per haploid genome, see below) to allow a consensus to be called from the data [20]. While short read Illumina data is unable to resolve long repeats, it has higher accuracy and can be used to correct long read data [4] either before or after assembly to improve the accuracy or completeness of the genome [20].

The recommended coverage for genome assembly varies from 40× to 60× at a minimum. For example, Li and Harkness [3] suggest 40–50× and Del Angel et al. [4] and Jung et al. [20] suggest a minimum of 60× for small, inbred, diploid genomes. Coverage is generally estimated based on the Lander-Waterman equation [92] as read length multiplied by read number divided by the haploid genome size for the species. Perhaps more simply for project planning, the amount of sequencing data needed for a project can be calculated by multiplying the estimated size of the plant’s haploid genome by the coverage needed. However, it is important to note that coverage will be reduced by quality control and filtering steps compared to the raw coverage. Additionally, the coverage will not be uniform across the nuclear genome. For example, up to 20% of the raw data may be DNA from the chloroplast resulting in relatively deep coverage of the relatively small chloroplast genome, but less coverage of the nuclear genome [93]. After generating sequence data, there are generally five, often iterative, steps before the “final” genome is ready for downstream analysis: 1) Data assessment and filtering, 2) assembly (often by multiple assemblers), 3) error correction and polishing, 4) scaffolding and/or the placement of scaffolds on chromosome sized pseudomolecules, and 5) annotation.

### 2.5. Data Assessment, Correction and Filtering

Before starting with the assembly process, it is advisable to assess the quality of the sequencing data and filter the reads based on this quality. However, some assemblers integrate quality filtering and correction as early steps in their assembly process and additional steps with alternative software may or may not improve the final assembly. Read length can also be a consideration as, for example, some long read assemblers will refuse to work if reads shorter than 500 bp are included in the input data. The software FastQC (www.bioinformatics.babraham.ac.uk/projects/fastqc/) provides a summary of quality parameters that is very helpful to assess the quality of short or long read data: Average per base quality, per tile quality, per sequence quality, per base content, per sequence GC content, per base N content, sequence length distribution, sequence duplication level, overrepresented sequences, adapter content and k-mer content. Overall quality of long read data can be also assessed with tools such as Nanoplot (github.com/wdecoster/NanoPlot) [94]. Correction of long read data with short reads can be done prior to assembly with tools such as LoRDEC (www.atgc-montpellier.fr/lordec) [95]. Filtering can be done with a variety of tools available online such as Trimmomatic (www.usadellab.org/cms/?page=trimmomatic) [96]. This type of software will generally remove reads or regions in the reads that are below a certain quality threshold as well as sequencing adapters or the “bar codes” of specific sequences that allow for identification of particular reads following multiplexing. Many custom scripts for filtering raw data can be found online (e.g., filter_fastq.py github.com/nanoporetech/fastq-filter/blob/master/filter_fastq.py). Users will want to apply the principle of *caveat emptor* when using these scripts, but they can provide invaluable tools.

### 2.6. Assembly and Assessment

Genome assemblers typically use either short or long read data as input. Short read assemblers have a longer history and many are designed with smaller bacterial or viral genomes in mind. However, because of their longer history, several of the programs that can handle larger genomes have also had extensive work to reduce the amount of computational resources they need such as ABySS 2.0 (www.bcgsc.ca/platform/bioinfo/software/abyss/releases/2.0.0) [97] and SOAPdenovo2 (github.com/aquaskyline/SOAPdenovo2) [98]. In our experience, two genome assemblers that use long read data that are relatively easy to install and use with strong documentation and community support are CANU (canu.readthedocs.io/en/latest) [99] and FALCON (pb-falcon.readthedocs.io/en/latest) [100]. CANU, in particular, appears to be a common choice (Table 1), perhaps because of the clarity of its documentation and recommendations on which parameters (e.g., correctedErrorRate and minOverlapLength) are the most likely to improve the outcome of the assembly. This type of guidance is very helpful as the key parameters for tuning software to a particular species are not always apparent, resulting in an overwhelming number of parameters that could be adjusted. However, when in doubt and lacking documentation, this information can also be gleaned from other users’ experience documented in discussion groups for the particular tool. Hybrid assemblers, that use both short and long read data, such as SPAdes (github.com/ablab/spades) [101], and Platanus-allee (platanus.bio.titech.ac.jp/platanus2 the recent replacement of Plantanus) [102] are available and assembly strategies that merge the results of multiple assemblers have also been used (e.g., [89]).

Once an assembler has completed a draft assembly of the genome, the challenge is determining how “good” the assembly is [19]. The definition of good can depend on the eventual use of the genome and includes parameters such as how contiguous (how many pieces is the genome in) the assembly is, how much of the genome was assembled and whether the assembly contains the expected genes. Often the first tool applied following genome assembly is QUAST (quast.sourceforge.net/quast), which provides a quick summary of the genome including the number of contigs, the total length of the genome as assembled, the N50, and, if the expected genome size is included the NG50 values. This gives an indication of contiguousness and the size of the assembly. BUSCO (busco.ezlab.org) [103,104] is frequently used as a quantitative measure of the completeness of a genome as it indicates whether the shared single copy genes expected in the genome are present—that is how much of the gene space has been captured and assembled. BUSCO indicates how many and which of these are complete and single copy, complete and duplicated, missing or fragmented (Table 1). Finally, BlobTools (blobtools.readme.io/docs) [101] can be used to determine if the assembled sequences are DNA from the expected organism or from contaminating organisms through taxonomic partitioning of the genome. This tool requires the draft genome sequence, a hit file created by BLASTn (blast.ncbi.nlm.nih.gov/Blast.cgi) [105] using the MegaBLAST option [106], a depth file created with a tool such as BWA-MEM (bio-bwa.sourceforge.net) [107], and the raw data used to assemble the genome sequence. After processing this information BlobTools creates a visual indication of which organisms are most closely related to the draft genome (Figure 2). If there is substantial contamination, this information to further filter the raw data for reassembly without the contaminating sequences.

### 2.7. Polishing

Polishing a genome can lead to significant improvements in the completeness of the genome as assessed by BUSCO and some tools will use short read data to call a consensus SNP, correct indels (insertions and deletions that are common in log read data) and misassembled contigs. Pilon (github.com/broadinstitute/pilon) [108] uses the assembled genome and one or more files containing the alignment of sequencing reads such as mate pairs, paired ends or unpaired sequences to the draft assembly. The program’s output includes the files needed for visualizing the changes to the genome using tools such as the Integrative Genomics Viewer (IGV software.broadinstitute.org/software/igv/) [109] and can generate information on the variation with genome sequence. PacBio has developed the tool GenomicConsensus (github.com/PacificBiosciences/GenomicConsensus), which uses mapped PacBio reads to generate a consensus, while Nanopolish (nanopolish.readthedocs.io/en/latest/index.html) has been developed for use with ONT data. In comparison, RACON (github.com/isovic/racon) can be used with either short read or long read data [110].

### 2.8. Scaffolding

Traditionally, the ordering and orientation of contigs into scaffolds has often relied on the labor intensive and expensive use of fluorescent in situ hybridization of bacterial artificial chromosomes (BACs) and segregating F2 populations that allow for mapping the position of the sequences. More recent methods: Chromosome conformation capture techniques (Hi-C), optical mapping techniques (Bionano) and 10x Genomics Chromium™ Systems can produce data that can be generated and applied to verify the assembly and generate scaffolds with less time and effort [3]. Chromosome conformation capture (3-C) has been a commonly used technique in molecular biology to map chromosomal interactions. It uses a process where genomic DNA is first digested and then ligated in conditions that preserve the 3D organization of the genome to allow the joining of distant sequences that find themselves to be in proximity. Using deep sequencing, the high throughput version of the technique (Hi-C) produces a genome-wide map of proximity contacts between all the different loci. Since the frequency of occurrence of such contacts is based on proximity, with intrachromosome contacts most common and the probability of contacts decreasing with distance, the technique can readily be used for scaffolding contigs [111]. If the analysis of this proximity data is not completed by the provider using proprietary software, once the paired end data has been mapped to assembled contigs, software such as SALSA (github.com/machinegun/SALSA) [112] can use the information to break misassembled contigs and scaffold the genome. FALCON-Phase (github.com/PacificBiosciences/pb-assembly) has also integrated the use of Hi-C data into the FALCON assembly pipeline through a collaboration between PacBio and Phase Genomics (www.phasegenomics.com) [113]. Phase Genomics is a USA based company that can provide kits for HI-C library preparation and bioinformatics support in the use of this data scaffolding of a *de novo* genome with their proprietary software Proximo. Additionally, they provide helpful advice on how to work with Hi-C data generated by their protocols (phasegenomics.github.io/2019/09/19/hic-alignment-and-qc.html). Recently, chromosome level assemblies of black raspberry (*Rubus occidentalis* L.) [114], an ornamental amaranth used by ancient civilizations in South and Central America as a grain crop (*Amaranthus hypochondriacus* L.) [115], and broomcorn millet (*Panicum miliaceum* L.) [59], genomes have been completed using Hi-C data and PacBio data.

Bionano Genomics (San Diego, CA, USA, bionanogenomics.com) contributes to scaffolding by optically mapping specific sequences distributed across the genome. Briefly, high molecular weight DNA is extracted, up to chromosome arm lengths, and labeled at specific sequence motifs for imaging and identification. The DNA molecule is then linearized onto a flowcell where a gradient of micro- and nano-structures gently unwinds and guides DNA into NanoChannels where it is imaged by a high resolution camera. The DNA fragments with similar motif-specific label patterns are assembled together to recreate a whole genome map assembly. This data can be used in a hybrid assembly to scaffold contigs obtained through sequencing of the genome. It can be used to identify regions that are incorrectly assembled or where structural variants can be found. This approach was recently used in the improvement of wheat’s hexaploid genome assembly [116] and the large *Sorghum* genome [117].

An alternative approach is used by 10x Genomics Chromium™ System (www.10xgenomics.com). DNA molecules are divided into small sets and provided with an identifying barcode before being sequenced. This provides linked reads that are unlikely to represent the same region from homologous chromosomes. This technique is particularly useful in genomes that are highly heterozygous and/or polyploid because it allows the genome information to be phased, that is the two haplotypes can be distinguished, and it can prevent the collapse of sequence from homologous chromosomes in polyploids. This technique was recently used in the sequencing of the octaploid strawberry genome (*Fragaria* X *ananassa*) [118].

An additional option when a related species with a chromosome-level genome sequence is available, is that this information can be used to create reference based assembly with chromosome-level resolution. However, this method would bias the assembly to more closely resemble that of the relative and will, for example, lack chromosome scale rearrangements. One option for pursuing this route, MeDuSa [119] (github.com/combogenomics/medusa/releases), can use one or more closely related genomes for generating a chromosome-level draft.

### 2.9. Gene Prediction and Annotation

Once a genome sequence of adequate quality has been produced, genes and other genetic elements such as transposons need to be identified. Gene prediction software such as AUGUSTUS (bioinf.uni-greifswald.de/augustus) [120,121] can be used to locate potential coding sequences along the genome sequence. This software has been improved over the years, starting from entirely ab initio gene prediction to include evidence-based discovery using expressed sequence tag (EST) sequences, RNASeq data (by way of hints) and with protein multiple sequence alignments. Repeated elements such as transposable elements (retrotransposons and DNA transposons), tandem or inverted repeats, can be located in the genome with software such as RepeatMasker (www.repeatmasker.org), RepeatFinder (www.cbcb.umd.edu/software/RepeatFinder) [122], or the recently developed Generic Repeat Finder (GRF) [123]. Additionally, there are a host of software packages and resources designed to detect and annotate specific types of transposable elements including SINE_scan (github.com/maohlzj/SINE_Scan) [124] for detected short interspersed nuclear elements (SINEs), the P-Mite database (pmite.hzau.edu.cn) [125] for finding miniature inverted-repeat transposable elements, and HelitronScanner (sourceforge.net/projects/helitronscanner) [126] for detecting helitrons—rolling circles that often capture gene sequences leading to gene duplication.

It is useful to know what the product of identified gene sequences code for and tools have been designed to assign gene ontology—information on a gene’s product’s molecular function, location and role (GO, geneontology.org) using standardized language. One of the most ubiquitous tools used is the basic local alignment search tool (BLAST) [105] in conjunction with the Genbank [34] databases to assign putative functions through shared identity or similarity of the translated gene product. Blast2Go (www.blast2go.com) [127] is a tool with a subscription fee that can automate this process. Free software packages are also available including the widely used Maker-P (www.yandell-lab.org/software/maker-p.html) [128] as pipeline designed to make the annotation of plant genomes more accessible to new groups and incorporates many of the software packages mentioned above and has extensive documentation and tutorials.

### 2.10. Examples: Three Recently Sequenced Weed Genomes

Given the wide variety of sequencing strategies and tools that can be employed (or not) at each stage of genome assembly it is unlikely that any two projects have followed the same path to a final assembly. Further, as noted by Del Angel et al. [4], it is important to set goals at the beginning of a project for how contiguous and complete the genome sequence needs to be for the specific project, otherwise the iterative process of analysis and reanalysis with alternative tools can be endless. Given the complexities of genomes (e.g., [129]) and how this complexity is reduced in a genome assembly, it may be helpful to consider a modification of George E. P. Box’s aphorism that all genome sequences are wrong, but some are useful. As examples of how these techniques and programs have been applied to weeds, we briefly summarize the methods and outcomes of three recent sequencing projects of two diploids, kochia (*Kochia scoparia* (L.) Schrad. also called *Bassia scoparia* (L.) A.J.Scott), common waterhemp (*Amaranthus tuberculatus* (Moq.) Sauer), and a hexaploid species, barnyard grass (*Echinochloa crus-galli* (L.) Beauv.).

For kochia, a plant with a genome size of approximately [89] 1Gbp (2n = 2x = 18), DNA for sequencing was extracted from a glyphosate susceptible inbred line using a modified CTAB protocol. They sequenced three Illumina libraries, one paired end and two mate-pair libraries using three HiSeq lanes and used 12 PacBio SMRT cells. They then assembled and merged two assembles into a final assembly for analysis. For the first assembly, they used the paired end data and the program Proovread (github.com/BioInf-Wuerzburg/proovread) [130] to correct the PacBio reads, which were then assembled with Canu. For the second assembly, ALLPATHS-LG (software.broadinstitute.org/allpaths-lg/blog) [131] was used to assemble all the Illumina data and scaffolding was completed using the PacBio reads and PBJelly (sourceforge.net/p/pb-jelly/wiki/Home) [132]. They then used the GARM Meta assembler (garm-meta-assem.sourceforge.net) [133] to merge the genomes. This final 711 Mbp assembly consisted of 19,671 scaffolds and had an N50 of 62 kb. Completeness as indicated by BUSCO, using the eudicotyledons odb10 dataset, was estimated at 70.3%. Kochia’s sequence was then annotated using the WQ-Maker pipeline transcriptome data from kochia and expressed sequence tags for kochia’s family, the Chenopodiaceae, from the National Center for Biotechnology Information (NCBI www.ncbi.nlm.nih.gov). Then then used BLASTN and BLASTP to predict genes and proteins and RepeatMasker to search for repetitive elements.

In the case of common waterhemp, a species with a genome size of approximately 676 Mbp (2n = 2x = 32), DNA from a single female plant was extracted using a modified CTAB protocol and sequenced with both PacBio reads, 15 SMRT cells, and one Illumina HiSeq lane of 150 bp paired end library reads [88]. The long read data provide 87× coverage and was assembled using Canu and then polished with the short read data using Arrow and Pilon. This resulted in a final genome assembly size of 663 Mbp consisting of 2,514 contigs and an N50 of 1.7Mb. The assembly contained 88% of BUSCO’s Embryophyta’s genes. The program REVEAL (github.com/jasperlinthorst/REVEAL) [134] was then used to produce 16 pseudomolecules using the chromosomal level genome assembly of the cereal crop species *Amaranthus hypochondriacus* L. Both this finished genome and the assembly used to create it were annotated using the MAKER pipeline (yandell-lab.org/software/maker.html) following identification and masking of repetitive elements with RepeatModeler and RepeatMasker.

Barnyard grass has an estimated genome size at 1.4 Gbp based on flow cytometry data and K-mer analysis [54] and a chromosome count of 2n = 6x = 54. DNA was extracted for sequencing from a plant collected from a rice paddy using a CTAB protocol. They sequenced the 48 SMRT cells of PacBio for long read data and both paired end and mate pair Illumina libraries using HiSeq runs. This level of sequencing effort resulted in 171× coverage of the genome. The short read data was assembled with SOAPdenovo2, scaffolded with OPERA-LG (sourceforge.net/p/operasf/wiki/The%20OPERA%20wiki) [135], and then gaps in this assembly were closed with GapCloser from SOAPdenovo2. The long read data was assembled with Canu and used to fill gaps in the short read assembly with PBJelly. The draft genome produced was 1.27 Gbp in length with an N50 of 1.8 Mbp. The authors used BUSCO and determined that 95.5% of the core eukaryotic genes were complete. RepeatModeler and RepeatMasker were used to find and mask repetitive elements. Then they used transcriptome data and three programs to predict genes GeneMark.hmm (exon.gatech.edu/GeneMark) [136], Fgenesh (www.softberry.com) [137], and AUGUSTUS.

## 3. Current Application: What Are Agricultural Weeds and Where Do They Come From?

Harlan and deWet defined weediness as “an adaptive syndrome which permits a species or variety to thrive and become abundant and difficult to eradicate within areas of human disturbance” [138]. Under this definition, crops are the result of intentional selection for vigor and fertility in the agricultural environment and weeds are the unintentional result [139]. A classic example of this is crop mimicry, where weeds have been selected by agricultural practices such as hand weeding to closely resemble a crop species [140]. This includes species such as false flax (*Camelina sativa* (L.) Crantz), which looks like, has similar time to maturity, and similar seed size to varieties of cultivated flax [141,142], and rice-mimicking varieties of barnyard grass [140]. A more pressing example is the evolution of HR (see Section 4) [143]. This second example illustrates, that as a group, weeds represent multiple independent origins of weediness and numerous examples of rapid adaptive evolution that present an opportunity not only to co-opt these adaptations for crop improvement or guide changes in agricultural practices to slow or thwart this evolution [144], but to provide fundamental insights into evolution [145]. Agricultural weed populations can be selected from populations adapted to natural disturbance regimes or from populations selected for these characteristics as crops, from populations of wild crop relatives, or from hybrids between the two [141,146,147,148]. Similarly, specific traits that contribute to adaptation to the agricultural environment, including alleles conferring HR, are selected within those populations. These origins and the loci underlying adaptive traits can be elucidated by examining genomic variation with weed populations.

### 3.1. Detecting the Signatures of Demographic Change and Selection on the Genome

Demographic and selective events change the patterns of variation across the genome, leaving a record of these processes. In weed populations, demographic and selective events may be closely intertwined as artificial selection from weed control measures can drastically change population size and composition. For example, weed populations might undergo rapid declines in population size (bottlenecks) resulting from herbicide application followed by population expansions after the evolution of HR, or the introgression of HR genes from one population into another. These processes can be difficult to disentangle from each other, as well as from patterns related to the variable recombination rate across the genome. However, demographic processes generally leave a signature across the entirety of the genome, while selection leaves a signal localized to the genes that confer higher fitness under the given environmental regime.

Over time, adaptation of a population to its specific environment and associated demographic events lead to divergence in allelic composition across the genome relative to other populations. This divergence leads to population structure and can be used to infer the past history of the sample, with populations sharing more similar allele frequencies more likely to share a recent evolutionary history. When a species exhibits population structure, we can assign individuals to recent common “ancestral populations” that can provide clues to their origin. This is often the basis of human ancestry assignment through home DNA tests, where your genotyping results are compared to the frequency of alleles across the globe to determine which geographic region contains the highest proportion alleles similar to those comprising your genotype [149,150]. Population structure can also provide evidence of hybridization and introgression when individuals show the signal of a mixed affinity to populations or species (admixtures). Again, this is similar to the assignment of percentage affiliation to different groups in human ancestry tests.

Population structure can be estimated at many hierarchical levels, from individual, to subpopulation, and across longer timescales at the phylogenetic level (e.g., STRUCTURE (web.stanford.edu/group/pritchardlab/structure.html) [151], AMOVA [152], and TREEMIX (bitbucket.org/nygcresearch/treemix/wiki/Home) [153]). While these methods aim to cluster individuals into discretely structured groupings, allele frequencies may instead continuously vary across space [154]. This may be especially likely for a recently expanded species due to serial bottlenecks and expansions, or along clines in latitudinal or environmental gradients where there is limited opportunity for long distance dispersal [155,156]. However, methods have been developed to test whether a population is more likely to showing continuous or discrete population structure [157]. In these cases, a model free approach such as principal component analysis may help to clarify population structure [158]. These data can also be used to infer past demographic processes using modelling approaches that allow estimation of parameters including ancestral population size, the number and timing of bottlenecks, time since divergence between populations, ancestral and contemporary levels of gene flow, and contemporary effective population sizes. Demographic modelling has been widely implemented to infer the history of sampled populations including δaδi (bitbucket.org/gutenkunstlab/dadi/src/master/) [159] and FastSimCoal (cmpg.unibe.ch/software/fastsimcoal2/) [160]. With genome-wide data from a population level sample, produced either through a reduced genome representation technique (see Section 4.3) or resequencing (sequencing of a genome of using less coverage and a template draft genome sequence) population structure and demographic history can easily be estimated through these variety of approaches discussed above to provide powerful insights into the source and origins of agricultural weed populations.

While genome wide information provides high resolution data on the distribution of allelic differences among samples due to demography, allelic differences due to selection can be inferred with care using integrative summary statistics and model based approaches. Currently, our understanding is that HR evolution often proceeds through drastic changes in allele frequency at the target gene—conveniently, a single locus of large effect provides the most power for detecting recent signals of selection and differentiating independent events. Three types of signal can be used to recognize selection: changes in allele frequencies (differentiation and diversity), patterns associated with linkage (homozygosity), and the pattern of nucleotide substitutions

First, regions near alleles selected by agricultural practices can be indicated by changes in allele frequencies. When a beneficial allele changes in frequency, becoming highly prevalent or fixed in a population sites nearby, linked to the selected allele due to a low probability of recombination, will show a depletion of genetic variation. The pattern resulting from the fixation of nearby neutral sites along with the selected site is termed a selective sweep [161,162,163,164]. An expectation following from this process is that the frequency of alleles under selection is expected to differ among populations experiencing different conditions (e.g., herbicide application or none) and this differentiation between populations is frequently expressed as Wright’s fixation index (F_ST_), though there are a host of related statistics [165,166]. If the F_ST_ of a locus is much larger than at other nearby or neutral loci, this can indicate positive selection.

Second, in addition to differentiation, immediately following selection the frequency of linked alleles will be fixed with new mutations causing new alleles to accrue slowly thereafter. This results in an excess of homozygosity (lack of variant sites) directly after selection. As new alleles will be rare, an excess of rare alleles can indicate positive selection (as well as recent population expansion) and can be quantified by Tajima’s D, which compares the number of pair-wise differences between individuals with the total number of segregating polymorphisms [167]. Similarly, Fay and Wu compare the number of pair-wise differences between individuals to the number of individuals that are homozygous for the allele [168].

Third, selection can be detected through a comparison of the rate of nonsynonymous substitutions at a nucleotide (those that alter the amino-acid represented by the codon) to the rate of synonymous substitutions, which are assumed to be silent and neutral. This ratio can indicate selection favoring a change in the structure of a protein (d_N_/d_S_).

Beyond these summary statistics, many model-based approaches have been developed to distinguish between recent, single genetic origin selective events (hard sweeps) and older or multiple genetic origin selective events (soft sweep) by assessing differences in the magnitude of their signals across the genome (e.g., SweeD (cme.h-its.org/exelixis/web/software/sweed/index.html) [169] and SweepFinder2 (www.personal.psu.edu/mxd60/sf2.html) [170,171]). After assaying within population sweep patterns, one can then compare the extent of convergence in these patterns across populations. A greater or lesser extent of parallel changes in allele frequencies, homozygosity, and diversity in the surrounding sequence provide evidence of shared or independent origins of resistance across populations respectively, and more broadly, may provide the means to identify candidate genes that appear to underlie HR in multiple populations (see Section 4).

While there is great potential to determine the source and number of independent and shared origins of HR from genomic data (e.g., [88]), the task will be more difficult when HR is conferred by many alleles of small effect. With polygenic trait architectures many individuals are needed to have sufficient power to detect the individual small-effect changes, and therefore approaches often rely on taking the sum of allele frequencies weighted by their effect size on the trait [172]. Since these genome-wide association approaches assume allele frequency differences across the genome are all related to selection, one must carefully account for allele frequency changes due to population structure, which has been shown to often be confounded with polygenic signals of selection [173].

### 3.2. Example: Convergent Adaptation to Glyphosate in Common Waterhemp

Common waterhemp is a problematic, a wind-pollinated, outcrossing, and dioecious weed that occurs throughout the mid-western and eastern United States of America and in Canada from Manitoba to Quebec. It has been hypothesized that weedy agriculture populations result from human-mediated disturbance and mixing of two closely related taxa, *A. tuberculatus* var. *rudis*, a Midwestern native, highly associated with agricultural environments, and *A. tuberculatus* var. *tuberculatus*, a species that occupies a constrained range, and that is limited to riparian environments [174]. Glyphosate resistance was first reported in 2005 in Missouri and one hypothesis is that it may have spread from there across the United States and recently into Ontario. However, considering the strength of selection from herbicides and the highly repetitive nature of HR evolution as suggested from independent glyphosate resistance evolution in multiple *Amaranthus* species [35], it is also possible that glyphosate resistance may have multiple independent origins with *A. tuberculatus*, representing a striking case of convergent evolution.

A recent study used genomic approaches to investigate the history of the species, clarify the origins of agricultural populations, and the evolution of glyphosate resistance [88]. Specifically, Kreiner et al. [88] sequenced the species’ genome as described above (see Section 2.6) and then resequenced the genomes of 163 individuals from 19 agricultural populations known to have glyphosate resistance, varying from 13% to 88% of the population, from Missouri, Illinois, and Essex County and Walpole Island within Ontario, as well as ten individuals from a native, non-agricultural population in Ontario that lacked glyphosate resistance. This data and the software freebayes (github.com/ekg/freebayes) [174] were used to identify SNPs across the species genome and then to characterize population demographics, diversity, differentiation, and structure. Demographic modeling completed using δaδi supported the hypothesis of recent secondary contact between lineages. Similarly, analysis with STRUCTURE and principal component analysis, indicated that populations were genetically differentiated by geography and hypothesized species ranges, with populations from Missouri and Illinois clustering and corresponding to *A. tuberculatus* var. *rudis* and natural populations from Ontario clustering and corresponding to *A. tuberculatus* var. *tuberculatus.* These analyses also showed resistant populations from Essex county were unlike nearby natural or agricultural populations found in Ontario, but rather clustered with western Missouri populations. This indicates that populations from Essex County likely represent an introduction of seed from Midwestern *A. tuberculatus* var. *rudis* populations, that harbored multiple independent resistance haplotypes. Interestingly, the second group of resistant populations in Ontario, those from Walpole Island, clustered with natural populations in the area, though with signs of some introgression from the var. *rudis* cluster. With information on the evolutionary origins of these populations, Kreiner et al. set out to distinguish whether populations with shared evolutionary origins have independently evolved resistance, or if resistance spread through the expansion of these populations into new agricultural landscapes. The authors investigated the pattern of selection on the chromosome bearing the glyphosate target-site gene, 5-enolpyruvylshikimate-3-phosphate synthase (EPSPS), using Sweepfinder2 and model-free summary statistics such as diversity, homozygosity, and differentiation. This analysis indicated the plants from Walpole showed a stronger pattern of reduced genetic diversity, increased differentiation and increased extended haplotype homozygosity around the EPSPS genes—evidence of a hard selective sweep—distinct from plants from Essex county, Missouri, or Illinois where a soft-sweep following multiple origins throughout the Midwest appears to have occurred. The authors conclude that glyphosate resistance in newly problematic Ontario populations has multiple genetic origins – both through new seed introduction events and selection on a recently arisen mutation in a previously benign population.

## 4. Current Application: What Genes Underlie Herbicide Resistance?

Understanding the genetic basis of resistance to an herbicide in a plant species is an essential first step in the development of diagnostic markers, understanding the fitness consequences of the mutation, and, more generally, in understanding how herbicide evolution typically occurs. This information is essential for being able to detect, monitor and develop more effective strategies for managing HR. Of the current total of 500 unique combinations of species (256) and herbicide site mode of action, the underlying genetic basis of these resistances is only known for a minority of cases [35]. The majority of known cases involve mutations to the herbicide’s target site (TSR), while the specific genetic basis of non-target site resistance (NTSR) is largely unknown [175,176].

Our lack of understanding of the genetic basis of NTSR, is a major gap in our understanding of weed biology and the evolution and spread of HR [175,176,177]. Non-target site resistance is the most common mechanism contributing to glyphosate and acetyl CoA carboxylase inhibition resistance (ACCase). It is also the most common mechanism for acetolactate synthase (ALS) resistance in grass species [178] and can confer resistance to several herbicide modes of action simultaneously and unpredictably [179]. Non-target site resistance encompasses a diverse and complex set of traits that likely involve the full gamete of potential genetic basis including dominant to semi-dominant alleles with major effects, copy number variation, multiple minor alleles that incrementally contribute to resistance, and changes in epigenetic regulation (reviewed by [180]). Further, NTSR likely involves varied aspects of the fundamental processes within cells from transcription to translation invoking complex stress responses and altering regulatory pathways [177,180,181]. Until this gap in our knowledge is filled in our ability to make diagnostic tests, draw conclusions about the type and prevalence of mutations/variation that contribute to HR or develop strategies to interfere with NTSR pathways is compromised. However, while we rarely know the specific genetic basis of NTSR in a weed species, we have a good understanding of the types of genes are most likely involved.

### 4.1. Five Superfamilies of Suspects

Five gene superfamilies have members that have been identified as likely involved in NTSR. Evidence for their involvement comes from either their ability to confer herbicide tolerance or resistance in crop species or *Arabidopsis*, on enzyme and transcriptome analyses of herbicide resistant species or investigations of the molecular mechanisms of drug resistance (reviewed by [182,183]). Evidence from transcriptome studies suggests NTSR is often the result of the action of multiple members of a superfamily and multiple superfamilies [184,185,186,187]. Each of these families are large, diverse, and widely represented across the tree of life from bacteria to mammals indicating that they are fundamental to how organisms cope with their environments. In this regard, the evolution of HR has selected variants of genes underlying the complex regulatory and enzymatic pathways that organisms have always used to face biotic and abiotic stresses [188]. These gene superfamilies are considered to form part of what has been termed the xenome, the chemical detection, transport and detoxification system of plants [189] and members of the families are spread throughout plant genomes.

#### 4.1.1. Cytochrome P450 Monooxygenases

The cytochrome P450 monooxygenase gene superfamily (CYP) are the largest enzyme family in plants and are known to be involved in HR [190]. This superfamily, which is involved in detoxification and stress responses, were implicated in HR as a result of the analysis of herbicide residues from plants, their induction following the application of safeners (chemicals that increase herbicide tolerance in grain crops), and the observation of increased P450 metabolism levels in HR annual ryegrass (*Lolium rigidium* Gaud.), black grass (*Alopecurus myosuroides* Huds.) and lesser canary grass (*Phalaris minor* Retz.) [177]. However, the number of these genes [191], with 272 in *Arabidopsis thaliana,* for example [192], and issues with purification from plant material meant that the isolation of specific CYP genes conferring HR in plants was preceded by isolation of these genes in bacteria and mammals, which frequently have higher activity than those from plants [193]. As an example, expression of human CYP genes in potato [194] and rice [195,196,197] confer HR. Indeed, expression of *CYP1A1* in rice resulted in resistance to ten different herbicides from ten different Herbicide Resistance Action Committee (HRAC) groups [195,198], while expression of *CYP2B6* in resistance to thirteen from six HRAC groups [197]. Despite this demonstrated ability of individual CYP genes to confer broad HR, it is likely that multiple CYP genes are involved in NSTR within each plant species [177]. Plant derived CYP genes that have been demonstrated to confer HR have now been isolated in Jerusalem artichoke (*Helianthus tuberosus* L.) [199], soybean (*Glycine max* (L.) Merr.) [200], *Arabidopsis* [201] and ginseng (*Panax ginseng* Mey.) [202]. Within weeds, two CYP genes have been determined to be associated with ALS resistance in rice barnyardgrass (*Echinochloa phyllopogon* (Staf).) Koso-Pol.) and overexpression of these genes in *Arabidopsis* resulted in resistance to group B herbicides bensulfuron-methyl and penoxsulam [203] and group F4 clomazone [204]. The isolation of CYP genes responsible for HR from other weed species will likely occur in the near future as chemical inhibition of P450 indicate that these genes are involved in HR for flixweed (*Descurainia sophia* L.) [205], water hemp (*Amaranthus tuberculatus* (Moq.) Sauer var. *rudis* (Sauer) Costea & Tardif) [206], and large crabgrass (*Digitaria sanguinalis* L. Scop.) [207] in addition to the grass species mentioned above. Additionally, consistent expansion of CYP copy number across all 69 annotated CYP genes in *Amaranthus tuberculatus* agricultural populations relative to natural populations has been recently found [88].

#### 4.1.2. Glutathione S-Transferases

Glutathione S-transferases (GSTs) are enzymes that play a strong role in plant secondary metabolism and stress response [208,209,210]. For example, GSTs have been identified as playing a role in salt tolerance [182], copper tolerance [211] and fungal disease resistance [212]. They were first identified in mammals in the 1960s because of their role in drug metabolism and their presence in plants was identified soon after as contributing to atrazine resistance in maize (*Zea mays* L.) [213]. As a result, the role of GSTs for herbicide detoxification in maize have been extensively studied [214] and several of the genes encoding these enzymes have been used to engineer HR. For example, *GST1* [215] expressed in tobacco (*Nicotiana tabacum* L.) [216], resulted in resistance to alachor (group K3) and *GST27*, when expressed in wheat (*Triticum aestivum* L.), resulted in atrazine (group C1) and oxyfluorfen (group E) resistance [217]. Similarly, overexpression of a GSTs from soybean, *GmGSTU4*, in tobacco results in a significant increase in alachor tolerance [218]. Within weeds, two glutathione-S-transferase genes have been identified as being involved in resistance to ACCase and ALS inhibitors in black grass [219]. Indeed, although multiple loci are believed to be involved in NTSR HR for black grass [220], expression of *AmGSTF1* in *Arabidopsis* resulted in resistance to atrazine, alachor, and chlorotoluron (group C2) [185]. Expression analysis suggests that GSTs are involved in HR for a number of other weed species including junglerice (*Echinochloa colona* (L.) Link.) [221], Palmer amaranth (*Amaranthus palmeri* S. Wats.) [222], annual ryegrass [184,223] and sunflower (*Helianthus annuus* L.) [224]. However, as with the CYP genes, the number of GSTs in a plants species makes pinpointing the specific gene or genes responsible for HR challenging. For example, there may be 42 in maize [225] and 54 functional GSTs have been identified in *Arabidopsis* [226].

#### 4.1.3. ATP-Binding Cassette Transporters

ATP-binding cassette (ABC) transporters are a group of proteins that mediate cross membrane transport (reviewed by [227,228]). With more than 80 members they are the largest protein family in *Escherichia coli.* Approximately 130 and 150 members have been located within the *Arabidopsis* [229] and the tomato (*Solanum lycopersicum* (L.) H. Karst.) [230] genomes, respectively. These transporters are understood to be involved in the transport of auxin and glyphosate and may, therefore, play a role when reduced translocation or sequestration of these herbicides is involved in HR [177,231]. In horseweed (*Conzya canadensis* (L.) Cronq.) glyphosate application caused increased expression level in at least seven ABC transporter genes [232] and a transcriptome study on the closely related hairy fleabane (*Conzya bonariensis* (L.) Cronq.) indicated that there were 19 ABC transporter genes in addition to 22 other candidates including GSTs and glycotransferases (see below). Additional evidence of the role of this group is that overexpression of the ABC transporter gene *AtPgp1* in *Arabidopsis* resulted in resistance to dicamba (group O) and oryzalin (group K1) [233] and tobacco overexpressing *pqrA* from the bacterium *Ochrobactrum anthropi* show higher resistance to paraquat (group D) [234].

#### 4.1.4. MFS Transporters

The major facilitator superfamily (MFS) are also transporter proteins. As with the ABC transporters, there are approximately 70 members of the family within the genome of *Escherichia coli* [235] with perhaps 200 in *Arabidopsis* [236]. Like the ABC transporters members of the MFS family have been identified as being upregulated following exposure to auxinic herbicides [237] and the *TPO1* gene from yeast is a member of this group and its homolog from *Arabidopsis, At5g13750,* are able to confer resistance to 2,4-D when overexpressed in yeast [238]. However, it does not appear that studies examining the consequences of over expression of this type of gene in plants have been completed.

#### 4.1.5. Glycosyltransferases

Glycosyltransferases (GTs), enzymes that add carbohydrates to molecules, are involved in the detoxification of herbicides in addition to many other roles within plant cells [239,240]. They are numerous in plant genomes with one particular family within this superfamily, the UDP-glucose dependent glycosyltransferases (UGTs), having 107 functional members in *Arabidopsis* [241]. Like CYP and GSTs genes, they are induced by the application of safeners and have been detected in transcriptome studies following herbicide application [189] and enzymes from this group from a wide variety of organisms have been demonstrated to have activity against atrazine and fluorodifen (group F1) [240]. However, unlike the other superfamilies discussed here, we did not find any examples of genes from this family being used to produce HR organisms. Instead, much of the work focused on these enzymes is examining the potential of these enzymes in phytoremediation of organic pollutants [189,242,243]. For example, a gene in *Arabidopsis* (*UGT72B1*) encodes an enzyme that detoxifies 3,4-dichloroaniline (DCA) and 2,4,5-trichlorophenol (TCP) [244].

### 4.2. A Role for Genomic Approaches

Due to the complexity, diversity, and number of genes that could underlie NTSR; identification of resistance-conferring mutations is a significant challenge even when one has a lead on the potential genetic basis from the above insights [180]. Clearly, significant progress is being made through the application of RNA sequencing to identify the genes being expressed following herbicide application, expression analysis of those genes using quantitative PCR, and transformation of model organisms such as *Arabidopsis* and tobacco to verify the function of the genes. Additional genomic information for weeds is an asset for this type of investigation and can allow comparative genetic approaches and searches with tools such as BLAST [106] to identify and classify members of the multigene families discussed above as has been done in model organisms and crops (e.g., [225]). This can allow for systematic testing of the activity each enzyme (e.g., [241,245]). However, there are undoubtedly more genes and gene families involved in NTSR (e.g., [246]). As with unravelling the demographic history and structure of populations discussed, one method of identifying these genes is to examine the signature of the strong artificial selection pressure of herbicide application across the genome (see Section 3.2). Additionally, a physical map combined with the tools of genetics (e.g., linkage mapping, genome-wide association studies) can inform on small to large effect genomic loci involved in HR.

### 4.3. Example: Glyphosate NTSR in Morning Glory

A recent *tour de force* investigating glyphosate resistance in morning glory (*Ipomoea purpurea* (L.) Roth.) provides a clear example of how genomics and detection of the signature of selection can be applied to understanding the basis of non-target site resistance. In this work, Van Etten and colleagues [187] generated genome wide DNA markers to examine population structure, the possibility of multiple origins of HR in the species, and to provide an indication of where selection was acting in the genomes. They then sequenced the species’ genome and re-sequenced targets within the exome, the regions of the genome that are the parts of a gene that encode the final RNA transcripts, in regions showing selection. This data was used to assemble multiple lines of evidence to identify the candidate genes underlying glyphosate resistance.

To provide information of population differentiation and structure, examine the evidence for HR genes being introduced to populations via gene flow versus the HR arising multiple times, and to search for signatures of selection Van Etten et al. [185] used a reduced genome representation technique (nextRAD). This approach identified single nucleotide polymorphisms (SNPs) across the species’ genome for ten individuals from each of four high and four low survival populations. This approach is a variant of restriction site associated DNA sequencing (RADseq), which in general, use restriction enzymes (often a pair) to selectively amplify regions adjacent to restriction sites across a species’ genome [247,248]. The number of markers can be manipulated through the length of the restriction enzyme’s recognition site allowing for the density of the markers to be manipulated depending on the project’s goal and species genome size. As no sequence data is required before hand, this type of data can be generated for species whether or not they have genome sequences available. For each individual, enough Illumina sequencing data needs to be completed to result in approximately 30× coverage for each amplified region. Then programs such as STACKS (catchenlab.life.illinois.edu/stacks) [249,250,251] or TASSEL (bitbucket.org/tasseladmin/tassel-5-source/wiki/Home) [252] can be used to either group reads by similarity, if a sequenced genome is unavailable, or to align the reads to a draft genome sequence to locate polymorphic (variable) SNPs. These SNPs can then be analyzed with a plethora of packages in the free statistical programing language R [253] to understand the population biology (reviewed by [254]). This can include calculation of population differentiation (F_ST_) using hierfstat [255] or StAMPP [256]; the generation and visualization of unweighted pair group method with arithmetic mean (UPGMA) or neighbor joining trees using poppr [257] and phytools [258]; and k-means clustering (adegenet [259]) to further investigate population structure. In the case of glyphosate resistance, in both morning glory [185] and Palmer amaranth (*Amaranthus palmeri* S. Wats.) [260], this approach indicated that gene flow introducing HR alleles has likely been responsible for much of the pattern of resistance and susceptible populations. However, in addition to gene flow, a second origin of glyphosate resistance was also suggested in Palmer amaranth [260].

The population level SNP data generated by Van Etten et al. [185] was then further analyzed with two programs, BayeScan [261], which can identify SNPs that show signs of selection and bayenv2 [262], which indicate SNPs associated with levels of HR. BayeScan (cmpg.unibe.ch/software/BayeScan/) calculates pairwise F_ST_ values between each population sampled and a theoretical population comprised of a common gene pool from all sampled populations. Selection is implied as an explanation, if a locus specific factor improves the logistic regression model for these F_ST_ values that includes population structure [261]. The program bayenv2 (bitbucket.org/tguenther/bayenv2_public/src/default/) looks for correlations between an environmental variable, such as HR level, and SNP frequency using a Bayesian method that estimates the pattern of covariance of allele frequencies, uses this as a null model and then tests each SNP [262]. Putative genes in proximity to the 42 outlier SNPs identified by BayeScan and the 83 SNPs flagged by bayenv2 were then identified by annotation tools such as AUGUSTUS (see above).

Next they sequenced a morning glory (diploid, approximately 978 Mb,1C = 1.0 pg [24], 2n = 30 [24])) individual that they considered to be high homozygous using PacBio reads (11 SMRT Cells) and Illumina short read data (100 bp paired end). They completed two genome assemblies one using only the Illumina data with the program ABYSS (github.com/bcgsc/abyss) [263] and the other using a hybrid approach that combined their long and short read data with the program DBG2OLC (github.com/yechengxi/DBG2OLC) [264]. This later assembly consisted of 17,897 scaffolds, had an N50 of 15,425 and a total length of 1,948 Mbp.

They then used their genome assembly to design probes (baits) to perform target-capture resequencing of these genes, the EPSPS genes, genes previously associated with HR and a randomly selected control group. This targeted exome re-sequencing was then completed for five individuals from each of their eight populations. These re-sequenced contigs were aligned to the chromosome level sequence of Japanese morning glory (*Ipomoea nil* (L.) Roth.) [265] to visualize the pattern of outliers indicating selection and they identified five regions of interest which contained 945 genes—including multiple members of the CYP, GSTs GT, and ABC transporter superfamilies. To determine if the number of members identified in these regions was greater than expectation for these large families, they resampled Japanese morning glory’s genome to provide a baseline estimate of the number of that would be expected. This indicated that GT, ABC transporters and CYP genes were each overrepresented in the identified regions. These five regions also showed high genetic differentiation between populations with high and low glyphosate survival.

One approximately 29 kb region aligned to Japanese morning glory’s chromosome 10 showed reduced nucleotide diversity in resistant individuals, strong evidence of selection based on Tajima’s D and Fay and Wu’s H as well as stronger linkage among the SNPs of this region. This region contained a tandemly repeated group of seven GT genes and nine CYP genes. For this region, they determined that the majority of resistant individuals shared high genetic similarity and tests of convergence suggesting that this region contains one or more beneficial genes that were introduced by gene flow and rapidly swept through resistant populations. While none of the non-synonymous SNPs in these genes showed fixation in the high survival populations, this region has a strong likelihood of containing loci that underlie glyphosate resistance in the species and are strong candidates for further functional validation.

## 5. Future Application: Can We Genetically Alter Weed Population to Make Them Easier to Control?

With a greater understanding of the population biology of weed species and the identification of the DNA sequence changes that underlie HR come opportunities for new control strategies. This is made particularly true by the development of genetic engineering methods involving clustered regularly interspaced short palindromic repeats (CRISPR) technologies. CRISPR tools are both simple and versatile, contributing to their successful spread in all aspects of molecular biology (reviewed in [266]). CRISPR systems are found in bacteria and archaea where they provide acquired immunity against invasive elements like phages. They do so by co-opting small pieces of DNA sequence from the pathogen which they subsequently use to generate guide RNA molecules that “program” an endonuclease (e.g., Cas9) to scan the genome and find its target. The recognition of DNA sequence homologous to the guide triggers the cleavage of the DNA strand that leads to mutations and potential inactivation of the targeted element.

In a landmark study, the CRISPR system of *Streptococcus pyogenes* was reduced to two components, an endonuclease (Cas9) and a single guide RNA, that could efficiently and specifically cut DNA in vitro [267]. Following this, similar two-component systems were introduced in a plethora of different organisms to engineer mutations in the DNA sequence with outstanding success [268]. Ultimately, the only requirement for this approach is the knowledge of the targeted DNA sequence, making application in weed control theoretically possible [269]. Consequently, while the short answer to the question “Can we genetically alter weed population to make them easier to control?” is *probably*, there are a great number of technical [270], ethical [271] and ecological [272] hurdles and no current examples of this approach being used in weed science. Here we focus on describing and discussing the potential and technical challenges to developing a weed control strategy using the engineering of whole populations. For an example, we reach beyond weed science to the control of insecticide resistant mosquitoes, summarizing the current findings and approaches of the scientists, who are likely to be the first to release gene drive element into the environment to control a pest population.

### 5.1. The Potential for Manipulation of Weed Populations

Ever since the demonstration of the repurposing of a bacterial CRISPR system as a programmable endonuclease [267], there has been speculation about its potential use for pest control or eradication [273]. Indeed, there were early successes in the application of CRISPR-based “gene drive” systems in order to decimate or modify populations of fruitfly (*Drosophila melanogaster* Meigen) and importantly, disease-spreading mosquitoes (*Anopheles stephensi* Liston and *Anopheles gambiae* Giles) (reviewed by [274]). The basis of a gene drive system relies on using a selfish genetic element capable of either copying itself or biasing reproduction towards its own inheritance so that it propagates through a population in a non-Mendelian fashion. This cheating of the classic inheritance rules can compensate for some deleterious consequences and potentially allow a measure of population control. Adding CRISPR components to this paradigm then allowed homing in on specific targets within the genomes making it available to newly sequenced weed plants [269]. Such a system has yet to be created in plants, but the rapid evolution of plant genetic engineering could make it a reality in the not too distant future. Indeed, in their report “Gene Drives on the Horizon” the National Academy of Sciences considers the potential of this strategy for the control of Palmer amaranth [271].

The overarching goal of such an endeavor is to create a transgenic weed able to introduce a genetic payload into the populations of its species using biased inheritance and resulted in populations that are easier to control because of a vulnerability introduced with the payload. What the ideal payload would be up for debate, but it is likely to include a CRISPR system composed of a gene encoding a programmable endonuclease like the *Streptococcus pyogenes* Rosenbach Cas9 and a single or multiple guide RNA. These guide RNA could be specifically designed to pair with the locus causing HR or, if this basis is unknown, the target locus could be unrelated to the HR allele, with the goal of introducing sensitivity to a new molecule altogether. The recognition of the target triggers catalytic activity and the cutting of the target DNA creating a lesion. Since DNA breaks are highly detrimental, they are quickly repaired by one of the many pathways existing in the host cell. The gene drive system then subverts the DNA repair pathways ensuring its own propagation. This step represents one of the major challenges to this approach, as plant cells are known to heavily favor non-homologous DNA repair pathways that only produce small DNA sequence changes [275] that would fail to propagate the selfish element.

Indeed, the success of gene drive methods in fruitflies and mosquitoes is due in large part to the frequent use of homology-guided DNA repair in insect cells. However, plant somatic cells seldom use homologous recombination and favor non-homologous repair mechanisms [275]. For gene drive elements to spread efficiently in a plant population, this ratio between the two types of repair would have to be altered. This would be critical as non-homologous repair would create alleles resistant to the CRISPR system that would counter efforts to spread the gene drive. This is why the precise insertion of the gene drive element at a chosen location in the weed genome will likely be a *sine qua non* condition to its propagation. Once integrated, the new allele can start competing with natural alleles, which it can target for cleavage and convert using the host cell machinery. Encouragingly, the molecular mechanism called gene targeting, which uses the same homologous host DNA repair pathways as the gene drive approach, is of great interest in plant genetic engineering and has greatly improved the past few years [276]. Gene targeting aims at delivering a DNA sequence of interest at a specific location within the genome and, therefore, has also greatly benefited from advances in CRISPR technologies. Just like gene drive, gene targeting requires the use of homology-guided DNA repair mechanisms instead of non-homologous DNA repair. The difference between the two is that the final goal of gene targeting is a single isolated event, while a gene drive must self-propagate indefinitely, thereby adding to the challenge.

Excitingly, the case of a bacterial transposon that co-opted a CRISPR system as a means to guide its own propagation within the genome was recently discovered [277]. Transposons are themselves selfish elements that have evolved different means to copy themselves to favor their propagation. For example, some transposons encode an enzyme called integrase that can insert a DNA fragment at a target site in a genome. This new molecular tool has enormous potential as a gene drive system being able to circumvent the need to coax the host repair machinery to use homologous repair mechanisms.

### 5.2. Additional Technical Challenges

There are a number of additional technical limitations in the creation of a useful gene drive system for weed management beyond a need for the target species to use homologous repair mechanisms. As a first hurdle, this approach would be restricted to plants that can be genetically transformed and little effort has been devoted to the development of transformation techniques in weeds. Plant susceptibility to transformation is highly variable and whether or not it is ultimately possible in a species depends on many intrinsic factors [278]. For instance, species with unfused carpels at the extremity of the stigma may be amenable to the convenient floral dip *Agrobacterium* mediated transformation method. However, the great majority of plant species relies on other methods, such as tissue culture with *Agrobacterium tumefaciens* Smith and Townsend or biolistic bombardment, both being much more time and resource consuming. It could, therefore, take a few months to many years to develop a new transformation protocol for a particular plant—a potentially sizable initial investment of resources.

When transformation is possible, the challenge of precisely integrating a given DNA construct remains. At the molecular level, the problem can be broken down into two distinct parts; the mobilization of the homologous repair machinery and the delivery of the DNA template to be copied in the genome. For the first part, it has been reported that expressing the CRISPR system in specialized cells where homology-guided DNA repair occur at higher frequencies can increase gene targeting [276]. We know for instance that cells undergoing meiosis rely on homologous recombination between DNA molecules for orchestrating proper chromosome segregation. One could take advantage of these cell-specific conditions and engineer a system that would only act in a specific cell context as was recently done in mouse female germline [279]. Another interesting avenue is the tethering of repair machinery components to the endonuclease. Indeed, the fusion of Cas9 with different proteins offers many opportunities including influencing downstream DNA repair as it was successfully done in human cells [280]. Such an approach could be tailored to improve the propagation success of a gene drive element. In the second part of the molecular cascade, a DNA template has to be provided for the homologous repair machinery to integrate at the break site. In the case of gene drive, the engineered allele would bear homology to the wild allele and would therefore present itself as a repair template. Interestingly, recent studies have shown increased success in gene targeting when using components of a geminivirus [281,282,283]. The rationale behind this approach is that viruses can generate multiple extrachromosomal copies of a given DNA sequence thereby increasing the chances of any one fragment being used as template by the repair machinery. This element could be included into a gene drive system to increase its efficiency.

### 5.3. Evolutionary Consequences and the Need for Integration with Other Management Strategies

Even without the numerous technical impediments to gene drive strategies in weeds, this approach presents enormous ethical, regulatory, and ecological challenges. Theoretically, a gene drive that reduces the fitness of a population or its ability to reproduce could bring a species to extinction, as it was convincingly demonstrated for caged mosquitoes [284]. Setting this as a goal seems unwise and unlikely to gain societal support [272,285,286] or regulatory approval [287], as a result, strategies to re-sensitized populations to an herbicide or create susceptible to a specific compound unlikely to be found beyond the agroecosystem are likely to be more tenable. The advantage of such an approach is that it does not reduce the fitness of the population in the wild per se. Like the use of herbicides, altering weed populations as a management strategy would not be a silver bullet and would require integration into integrative weed management strategies. In part, this would be a consequence of the time needed for alleles to spread through populations as this could take 10 to 20 generations for a gene drive system to saturate a population [288]. In the re-sensitizing approach, this would mean forsaking the use of a given herbicide for many years thereby relying on other control strategies. In this regard, creating a susceptibility to a new molecule would present advantages but great care would need to be taken in choosing such a compound.

A second reason why this strategy would need to be part of an integrated weed management strategy, comes from the lesson we have learned from our reliance on herbicides. Plants are quite able to evolve in response to selection through modification of genetic machinery, the exome (see Section 4), and the biotic challenge represented by a gene drive element will result in selection on similar genetic machinery used to counter similar genetic attacks from viruses or selfish genetic elements. For example, in the case of a CRISPR-based gene drive, any synonymous mutation to the targeted site(s) would severely reduce the efficiency of the endonucleolytic cleavage [289]. This has already been demonstrated in model species such as fruitflies [290]. The emergence of such allele would be expected and could be mitigated by selecting sites where mutation would have high fitness cost would be more likely to provide a robust solution [291]. Since CRISPR genes come from bacteria, there is also a chance the plant cell would silence them using intrinsic mechanisms and a silenced allele could render then organism “immune” to the subsequent use of a CRISPR-based approach. Taken together, all these considerations argue for thorough modelling and confined population studies before such a strategy could be released in the fields as has been laid out in recommendations by the National Academy of Sciences [271].

### 5.4. Example: Gene Drive in Malaria Vector Mosquitos

While examples of gene drive development in weed species remain for future reviews, significant work has focused on using the technology to control mosquitoes that spread malaria. This is a system with parallel challenges to those faced in weed science including the emergence of multiple-insecticide resistance with both target site and NTSR mechanisms and a lack of new chemical control options [292,293]. Malaria is a serious and prevalent disease with over 200 million cases a year. It is often fatal, particularly in children, and disproportionally affects people living in South America, South Asia and sub-Saharan Africa where access to health care is often limited. The World Health Organization reported that of the 435,000 deaths reported in 2017 from malaria, ninety-two percent occurred in Africa and sixty-two percent occurred in children under five [294]. Malaria can be caused by any one of five *Plasmodium* parasites and can be transferred by several of the 450 species of *Anopheles* mosquitoes [294]. Within the sub-Saharan Africa region, malaria is primarily the result of infection by *Plasmodium falciparum* Welch transferred by female *Anopheles gambiae* mosquitoes [294]. Chemical strategies for controlling populations of these mosquitoes have resulted in the evolution of insecticide resistance with the first cases of pyrethroid resistance reported in Sudan in the 1970s and reports of resistance now available across Africa and in Madagascar [295]. Currently, *A. gambiae* populations in regions such as the Côte d’Ivoire and Burkina Faso, have evolved complete resistance to all approved classes of insecticides [296,297]. In 2015, researchers developed a CRISPR-based gene drive system designed to reduce reproductive capability by disrupting the sequence of a gene likely involved in the development of the embryo’s body plan which results in female sterility. When carriers of this this gene were crossed to wild type mosquitoes the gene had a transmission rate of just over 99% and it was able to spread through a caged populations initiated from equal numbers of wild type and transformed individuals [298]. However, nuclease-resistant variants that completely blocked the spread of the gene could be detected as early as the second generation [285]. More recently, in 2018, the researchers improved on these results by disruption of a gene that controls sex differentiation and that has alternative splicing patterns in male and female mosquitoes, a characteristic believed to increase the constraints in the development of resistant variants. One of the two cages, initiated with 12.5% disrupted allele frequency, reached 100% allele frequency at generation 7 and extinction at generation 8, while for the second cage these two points were reached at generation 11 and 12 respectively. Importantly, they did not detect an evidence for the evolution of resistance to this gene drive, though they note that it may not be “resistance-proof” given a wider sample of mutations [284]. This work relied on foundational genomic information from *A. gambiae*’s genome sequence in 2002 [299] as well as detailed knowledge of the genetic basis of fundamental aspects of *A. gambia*’s biology. In July 2019, the researchers initiated small scale releases of genetically modified, sterile males (not equipped with gene drive) in Burkina Faso to produce the data required to meet the ultimate goal of releasing individuals with gene drive to control malaria [300]. The researchers that have developed this technology work with a consortium, Target Malaria (targetmalaria.org), that includes scientists, regulators, and community engagement specialists. They have also worked to understand the ecological risks associated with the unconfined release of this event [301]. This approach to develop the social license and regulatory approval for this type of intervention provides a valuable template for how weed scientists could approach the modification of a weed species for population management.

## 6. Conclusions

Genomic approaches are extremely powerful tools for understanding biological systems. These tools, while currently underutilized in weed biology, are exciting in their potential to answer key weed science questions and increasingly accessible. Here our goal is to provide a foothold for weed scientists considering this type of research by providing an introduction to the considerations and process of creating a draft genome and illustrating how that genome could be used as a fundamental tool. Draft weed genomes can provide a resource for demographic analyses that examine the result of selection on the genome. This information can shed light on the evolutionary origins of weeds allowing us to identify management practices that could prevent HR evolution. It can identify strengths and weaknesses of weed populations that can be targeted for control, while providing fundamental information on how plants rapidly respond to selection from humans. The changes that selection makes to the genome and revealed by genomic approaches can also provide evidence of which loci are the genetic basis of NTSR. This information will allow us to form strategies to interfere with these HR mechanisms. Finally, the insights we gain from a better understanding of weed species at the population, genomic and genic level using these approaches open the option of altering the genome of weed species to provide us another tool for weed management—a strategy nearing implementation in mice and mosquitoes.

## Figures and Tables

**Figure 1 plants-08-00354-f001:**
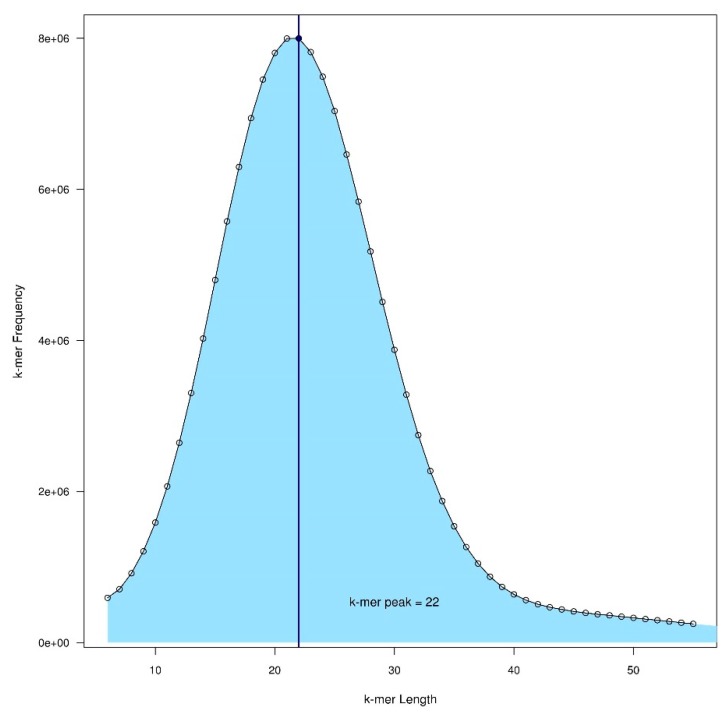
Plot of k-mer frequency by length produced for *Camelina neglecta* J.Brock, Mandáková, Lysak & Al-Shehbaz produced using Jellyfish and visualized using R. The position of the peak at a k-mer length of 22 is used to calculate genome size based on the area under the curve as represented by the light blue region. Here the genome size estimated is 248 Mb, while flow cytometry estimates indicate a genome size of 264 (±9) Mbp [80].

**Figure 2 plants-08-00354-f002:**
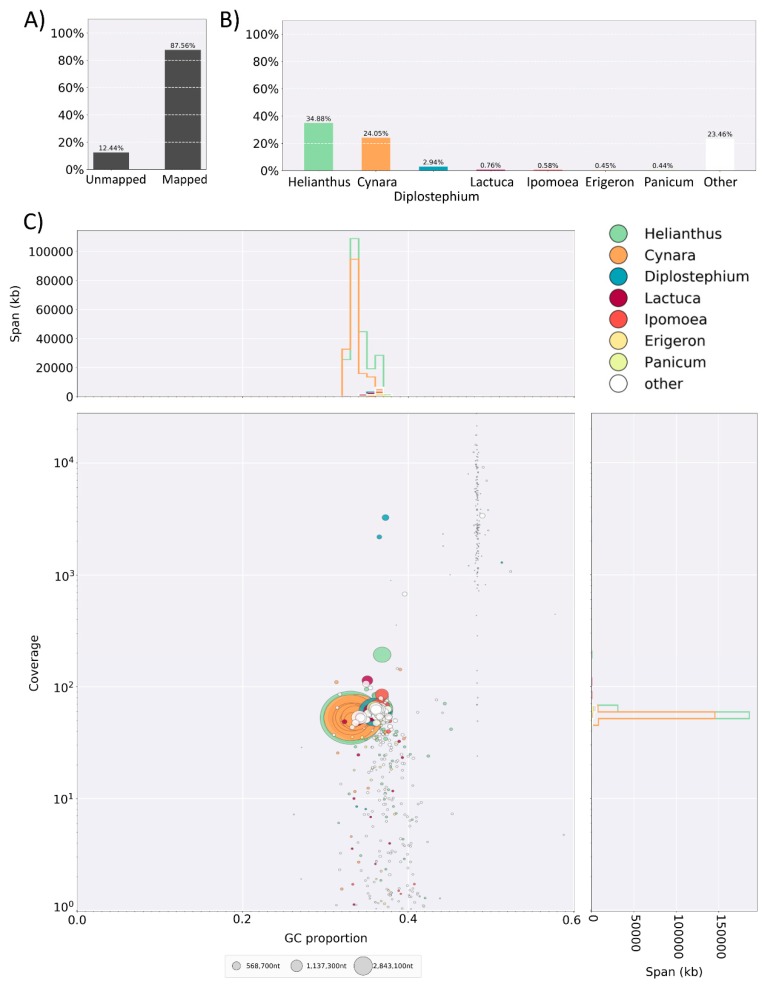
Blobplot generated for *Conzya canadensis* (Asteraceae) draft genome assembly showing the genera with the closest similarity to the sequenced genome (Laforest, Martin, and Page unpublished data). The first panel (**A**) indicates the percentage of reads that were mapped and the second panel (**B**) shows the taxonomic break down of hits at the taxonomic level requested. In this case the majority of hits are from other genera from the Asteraceae. The program generates a text file with more detailed information. The three part third panel (**C**) shows histograms for the proportion of G and C bases in the sequence which typically varies among species (top) and coverage (right) weighted by the cumulative length of sequences in each bin. The main panel has circles colored by taxonomic affiliation positioned on the x-axis by the GC proportion and on the y-axis by coverage within the raw data which gives a sense of the relative concentration of the sequences in the DNA sample.

**Table 1 plants-08-00354-t001:** Metrics of continuity and completion for weed genome assemblies available from GenBank. This list was compiled by search GenBank [34] in May 2019 for species included on one of the following five lists of weeds: 1) Species with herbicide resistance maintained at weedscience.org by Heap [35], 2) the United States Department of Agriculture’s Federal Noxious Weed List [36], 3) Weeds of Nation Significance in Australia [37], 4) Weber and Gut’s list of weeds spreading in Europe [38], or 5) the Canadian Weed Seed Order [39]. Year is the year the assembly was submitted to GenBank, and as with assembly level, coverage, sequencing technology used and assembly method were recorded from the assembly information page on GenBank. The number of contigs (greater than 500 bp long), assembled genome size, N50 and NG50 were determined by QUAST (v. 5.0.2). The number of BUSCOs that were (C)omplete, complete and (S)ingle-copy, complete and (D)uplicated, (F)ragmented, or (M)issing were determined using the eudicotyledons_odb10 set of 2121 conserved genes and BUSCO version 3.0.2. Where we could not locate a published for the genome, we have reported the lead author as listed as having submitted the genome to GenBank. Note that additional weed genomes may be available on CoGe, Phytozome, and the European nucleotide database.

Common Name	Latin Name	Year	Level of Assembly	No. of Contigs	Est. Genome Size (Mbp)	Assembled Size (Mbp)	N50	NG50	BUSCOS (Percentage of 2121 Genes)					Coverage	Sequencing Technology	Assembly Method	Reference or Lead Submitting Author
									C	S	D	F	M				
**Milkweed**	*Asclepias syriaca*	2017	Scaffold	221,885	411 ^1^	237	2555	NA^2^	76	75	1	13	11	80.4	Illumina	Platanus SCUBAT	[40]
**Winter Cress**	*Barbarea vulgaris*	2016	Scaffold	7810	270	167	56,351	19,454	95	93	2	3	3	66.5	Illumina	Celera	[41]
**Japanese Barberry**	*Berberis thunbergii*	2018	Contig	11,815	1515 ^1^	2241	397,058	654,137	88	30	57	3	9	104.8	PacBio	FALCON-Unzip	R. Bartaula
**False Brome**	*Brachypodium distachyon*	2018	Chromosome	11	355	271	59,130,575	59,130,575	80	76	4	6	15	9.4	ABI 3739	ARACHNE	[42]
**Bird Rape**	*Brassica rapa*	2017	Scaffold	70,673	485	386	3,737,062	2,395,810	98	80	18	1	1	212	Illumina PacBio	SOAPdenovo	[43]
**Hemp**	*Cannabis sativa*	2018	Chromosome	6653 ^3^	820	892	60,968,100	62,039,859	88	72	16	4	7	79	PacBio	FALCON	[44]
**Shepherd’s Purse**	*Capsella bursa-pastoris*	2017	Scaffold	8186	391 ^1^	268	627,605	320,701	96	13	83	2	3	40	Illumina	Newbler Platanus	[45]
**Horsetail Sheoak**	*Casuarina equisetifolia* subsp. *incana*	2018	Scaffold	2936	340 ^1^	301	1,020,118	894,734	97	93	4	1	2	546.9	Illumina PacBio	SOAPdenovo2 FALCON DISCOVAR	[46]
**Swamp Oak**	*Cauarina glauca*	2018	Scaffold	39,787	340	283	964,272	627,004	97	93	5	1	2	890	Illumina	SOAPdenovo	[47]
**Mandarin Orange**	*Citrus reticulata*	2018	Scaffold	67,725	460	344	1,376,405	577,147	98	96	2	1	1	200	Illumina	Platanus	[48]
**Horseweed**	*Conyza canadensis*	2014	Contig	20,075	335	326	20,748	20,226	66	44	22	10	24	350	Roche 454 Illumina PacBio	Newbler SOAPdenovo CLC NGS Cell	[49]
**Jute Mallow**	*Corchorus olitorius*	2017	Contig	52,373	450	377	16,573	13,050	93	90	3	3	4	47.7	Illumina	Newbler	[50]
**Muskmelon**	*Cucumis melo*	2012	Scaffold	10,823	450	375	4,428,067	3,741,400	94	92	2	2	4	13.5	Roche 454 Illumina	Newbler	[51]
**Globe artichoke**	*Cynara cardunculus*	2018	Chromosome	8283 ^3^	1084	725	25,947,084	173,700	96	90	6	2	2	80	Illumina	AllPaths	[52]
**Orchardgrass**	*Dactylis glomerata*	2018	Scaffold	1,072,009	3327 ^4^	840	3242	NA^2^	76	72	4	8	16	50	Illumina	SOAPdenovo	J. Li
**Carrot**	*Daucus carota* subsp. *sativus*	2016	Chromosome	4826 ^3^	473	422	36,610,139	36,610,139	94	88	6	2	5	186	Roche 454 Illumina Sanger	SOAPdenovo GapCloser	[53]
**Guinea yam**	*Dioscorea rotundata*	2017	Chromosome	21	694 ^1^	457	25,272,979	NA^2^	83	78	5	3	14	100	Illumina	Allpaths-LG SSPACE Premium	S. Natsume
**Barnyardgrass**	*Echinochloa crus-galli*	2017	Scaffold	4113	1400	486	705,200	NA^2^	89	26	63	2	10	170	Illumina PacBio	SOAPdenovo2 CANU	[54]
**Paterson’s curse**	*Echium plantagineum*	2019	Chromosome	1091 ^3^	333 ^1^	349	1,429,328	1,517,519	96	46	50	1	3	115	Illumina PacBio	MECAT LACHESIS	C.-Y. Tang
**Common sunflower**	*Helianthus annuus*	2017	Chromosome	1528 ^3^	3600	3028	178,899,001	174,509,413	89	80	9	3	8	100	PacBio	PBcR	[55]
**Littlebell**	*Ipomoea triloba*	2018	Chromosome	16	496 ^1^	462	29,809,665	28,894,297	97	89	7	1	2	290	Illumina PacBio	SOAPdenovo2 SSPACE PBJelly, Pilon	[56]
**Perennial Ryegrass**	*Lolium perenne*	2016	Scaffold	666,180	2621 ^1^	481	1361	NA^2^	31	29	2	22	47	5	Illumina	CLC Genomic Workbench	[57]
**Horsemint**	*Mentha longifolia*	2016	Scaffold	190,876	400	353	3915	3044	58	52	5	20	22	33	Illumina PacBio	MaSuRCA	[58]
**Amur silver grass**	*Miscanthus sacchariflorus*	2018	Chromosome	105,321 ^3^	2513 ^1^	2075	37,709	24,189	49	41	8	17	33	60	Illumina	ABySS SOAPdenovo2	J. De Vega
**Longstamen Rice**	*Oryza longistaminata*	2014	Scaffold	9688	782 ^1^	362	30,401,905	NA^2^	86	80	6	4	10	52.5	Illumina	SOAPdenovo2	C. Brian
**Red Rice**	*Oryza punctata*	2014	Chromosome	12	586 ^1^	394	31,244,610	28,494,620	81	74	7	6	13	130	Roche 454 Illumina	AllPaths	R. A. Wing
**Brownbeard Rice**	*Oryza rufipogon*	2015	Scaffold	3818	450 ^1^	339	27,785,585	26,200,591	83	76	6	5	13	120			Q. Zhao
**Rice**	*Oryza sativa*	2019	Chromosome	367 ^3^	489 ^1^	415	28,085,715	26,003,091	88	81	6	3	9	148	PacBio	CANU	L. Wang
**Broomcorn Millet**	*Panicum miliaceum*	2018	Chromosome	466	923	848	48,259,421	45,112,342	83	25	58	4	13	160	Illumina PacBio	CANU	[59]
**Opium Poppy**	*Papaver somniferum*	2018	Chromosome	34,381 ^3^	2870	2716	204,470,928	180,516,484	95	29	65	1	4	239	Illumina PacBio ONT	DeNovoMAGIC FALCON	[60]
**White Poplar**	*Populus alba*	2019	Contig	6087	508 ^1^	707	248,703	390,844	95	52	43	1	3	130	Illumina PacBio	SMARTdenovo	[61]
**Algarrobo blanco**	*Prosopis alba*	2019	Contig	4454	391 ^1^	500	237,044	357,710	70	49	21	3	27	30	PacBio	CANU	W. Kong
**Wild Radish**	*Raphanus raphistrum*	2014	Contig	64,732	515	254	10,333	NA^2^	95	82	12	3	2	47	Roche 454 Illumina	ABySS, Newbler, Celera Assembler, Minimus2	[62]
**Radish**	*Raphanus sativa*	2017	Chromosome	44,239 ^3^	573	383	35,166,889	26,198,371	96	82	14	2	1	225	Illumina	SOAPdenovo2	[63]
**Japanese Rose**	*Rosa multiflora*	2017	Scaffold	83,189	711	740	90,830	95,085	91	66	25	4	5	327	Illumina	SOAPdenovo2 GapCloser	[64]
**Wild Sugarcane**	*Saccharum spontaneum*	2018	Chromosome	15,303 ^3^	1565 ^1^	3133	91,359,291	109,189,819	78	20	58	5	17	90	Illumina PacBio	CANU HiC	[65]
**Rye**	*Secale cerale*	2017	Scaffold	1,581,707	7900	1685	2200	NA^2^	66	62	4	13	21	50	Illumina	CLC Assembly Cell, CarmA	[66]
**Green Foxtail**	*Setaria viridis*	2019	Chromosome	14	782 ^1^	396	46,702,114	35,460,007	81	75	6	6	13	118	PacBio	MECAT	P. Huang
**White Campion**	*Silene latifolia*	2018	Scaffold	319,506	2640 ^1^	1185	11,019	NA^2^	68	66	4	13	18	40	PacBio	SOAPdenovo2, CLC, PBJelly, SSPACE	[67]
**Milk Thistle**	*Silybum marianum*	2016	Contig	258,575	792 ^1^	1478	6967	NA^2^	38	33	6	8	54	96	Illumina PacBio	Celera Assembler	Y. Lv
**Sorghum**	*Sorghum bicolor*	2017	Chromosome	869 ^3^	730	709	68,658,214	68,658,214	86	80	5	4	10	8	Illumina Sanger	ARACHNE	[68]
**Stinkweed**	*Thlapsi arvense*	2015	Scaffold	6768	539	343	140,815	NA^2^	98	97	2	1	1	80	Illumina PacBio	CLC NGS Cell	[69]

^1^ When not reported by the authors, we have estimated based the genome size based on the genome size available from Kew’s C-DNA value database (see Section 2.2). In some cases, this has resulted in an estimate smaller than the assembled genome size. ^2^ In cases where the genome assemble size is not sufficiently higher than half of the expected genome size, an NG50 cannot be calculated (see Section 2.1). ^3^ In some cases chromosome-level genome assemblies have pieces left over and these increase the number of contigs included in the assembly files beyond the expected chromosome number. ^4^ Genome size estimate from DNA content analysis in Creber et al. [70].

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
