# Peer review of "Population Genomic Approaches for Weed Science"

_plants, 2019, doi:10.3390/plants8090354_

Round 1
Reviewer 1 Report
The topic of this manuscript is very interesting. This review provided an updated overview on genomic approaches for weed science. I consider that the manuscript contains information that deserve to be published after major revision.
I provide below a few suggestions that, if the authors decide to implement into the paper, the paper will improved.
Comments
Abstract: This section should be revised. The conclusions section is better written.
Introduction section: This section is too short. More information about genomic approaches for weed science should be provided.
Section 2. Developing draft weed genomes as a fundamental tool.
This section should be limited since there is not the main section of this review article. Also, authors used a lot of website references. These references should be limited.
In table 1, the correct latin name of stinkweed is Thlapsi arvense.
Section 3. What are agriculture weed and where do they come from?
This section should be revised. Authors should give more examples about specific weed species.
Section 4. This section is well written.
Section 5. Can we genetically alter weed population to make them easier to control.?
Authors give several examples for fungi and insects while the examples about weed species are very few. The question was not fully answered.
Author Response
We thank the reviewer for their comments.
Comment: Abstract: This section should be revised. The conclusions section is better written.
We have revised the abstract.
Comment: This section is too short. More information about genomic approaches for weed science should be provided.
In the introduction we lay out our general motivation and indicate the genomic approaches for weed science that that we will cover in detail in the body of the paper, rather than discussing them in detail in the introduction itself. Unless there are specific things that the reviewer can indicate that they feel are missing from the section, it is hard to revise accordingly.
Comment: Section 2. Developing draft weed genomes as a fundamental tool. This section should be limited since there is not the main section of this review article. Also, authors used a lot of website references. These references should be limited.
We have revised this section to make it shorter, however we feel the a large part of the motivation of this review paper is to provide detailed information on how a draft genome is produced as part of the orientation of weed scientists to the area. While that we agree that often websites are not always suitable as references for the scientific literature, they are needed and appropriate in this case as they typically point the reader to specific tools that maybe difficult to locate with Google or other search engines.
Comment: In table 1, the correct latin name of stinkweed is Thlapsi arvense.
Latin name corrected.
Comment: Section 3. What are agriculture weed and where do they come from? This section should be revised. Authors should give more examples about specific weed species
In this section we provide three specific examples. Indeed, a quarter of the section is dedicated to discussing one of these specific examples and the cited references contain numerous additional examples.
Comment: Section 5. Can we genetically alter weed population to make them easier to control.?
Authors give several examples for fungi and insects while the examples about weed species are very few. The question was not fully answered.
We have revised this section to make it clearer that the section is speculative in nature and that, while –no– examples exist for herbicide resistant plants, there is an example with strong parallels in insecticide resistant mosquitoes that is nearing implementation. This example including issues with ethics, social licence, ecological and evolutionary considerations and is highly relevant to the informed and considered discussion that should occur in the weed science community before the implementation of a similar solution.
Reviewer 2 Report
This paper deserves for publication. However, the authors should provide the most significant results from their paper, not only questions that need to be resolved in the Abstract part. Then it can be accepted.
1. Provide more details about what they discussed, for instance details of genes resistant to herbicides and advance of gene technology can be applied in weed management.
2. Revise errors, eg 4.4.1 and 4.1.2 to 4.1.5, I think need to be italic.
3. Line 489, L.., so erase .
4. Erase the part 5.4, I do not understand the relation of this research topic and this part.
5. Add a part of biological control of weeds using gene technologies. For instance, the determination of genes determined momilactones A and B in rice plant, which can be used to control barnyardgrass, monochoria and other weeds through plant breeding. Add recent papers related to momilactones A, B and momilactone-like compounds for weed control which has recently published in Plants-Basel.
6. Shorten the part 2.2-2.10 for more concisely, it was too long with many unneeded information for a review.
7. Add more information about the role of secondary metabolites for weed control and relation with relevant genes in weeds and crops.
8. Figure 1 is not needed for this review.
Author Response
We thank the reviewer for their comments.
1 . Provide more details about what they discussed, for instance details of genes resistant to herbicides and advance of gene technology can be applied in weed management.
We are unclear what the reviewer means here. This review is about genomic approaches as a tool for understanding weed populations. While this could include understanding more about specific gene families or detecting the putatively causative alleles for HR (Section 4), it is not intended to be about specific genes that can be incorporated into crops to provide HR. This would make an interesting subject for a different review paper.
2. Revise errors, eg 4.4.1 and 4.1.2 to 4.1.5, I think need to be italic.
Subsections are now in italics. Sub-subsections remain in normal type face as seems to be indicated in the template. Perhaps the editor can advise, if this requires alteration.
3. Line 489, L.., so erase .
Done.
4. Erase the part 5.4, I do not understand the relation of this research topic and this part.
We have revised this section to make it clearer that the section is speculative in nature and that, while –no– examples exist for herbicide resistant plants, there is an example with strong parallels in insecticide resistant mosquitoes that is nearing implementation. This example, including issues with ethics, social licence, ecological and evolutionary considerations and is highly relevant to the informed and considered discussion that should occur in the weed science community before the implementation of a similar solution.
5. Add a part of biological control of weeds using gene technologies. For instance, the determination of genes determined momilactones A and B in rice plant, which can be used to control barnyardgrass, monochoria and other weeds through plant breeding. Add recent papers related to momilactones A, B and momilactone-like compounds for weed control which has recently published in Plants-Basel.
Specific genes that may aid in weed management if used to transform crop species, such as the momilactones A and B in rice plants, increased stature, increased vigour etc.. are beyond the scope of this review (see 1). We look forward to the reviewer’s review paper on this alternative subject.
6. Shorten the part 2.2-2.10 for more concisely, it was too long with many unneeded information for a review.
We have revised this section to make it shorter, however we feel the a large part of the motivation of this review paper is to provide detailed information on how a draft genome is produced as part of the orientation of weed scientists to the area.
7. Add more information about the role of secondary metabolites for weed control and relation with relevant genes in weeds and crops.
Again, this is not in scope of this review paper. See points 1 and 5.
8. Figure 1 is not needed for this review.
Deleted
Round 2
Reviewer 1 Report
The authors following the comments during the reviewing process improved the manuscript. Thus, this article can be accepted for publication on this journal.